# Dupilumab in the Treatment of Severe Uncontrolled Chronic Rhinosinusitis with Nasal Polyps (CRSwNP) and Comorbid Asthma—A Multidisciplinary Monocentric Real-Life Study

**DOI:** 10.3390/biomedicines13020501

**Published:** 2025-02-17

**Authors:** Gian Luca Fadda, Chiara Rustichelli, Simone Soccal, Simone Moglio, Alessandro Serrone, Francesca Bertolini, Vitina Carriero, Stefano Pizzimenti, Stefano Levra, Giovanni Cavallo, Fabio Luigi Massimo Ricciardolo, Giuseppe Guida

**Affiliations:** 1Department of Otorhinolaryngology, University of Turin, “San Luigi Gonzaga” Hospital, Regione Gonzole 10, Orbassano, 10043 Turin, Italy; dott.fadda@gmail.com (G.L.F.); chiararustichelli75@gmail.com (C.R.); simone.moglio@unito.it (S.M.); alessandro.serrone@unito.it (A.S.); giovanni.cavallo@unito.it (G.C.); 2Department of Clinical and Biological Sciences, University of Turin, Orbassano, 10043 Turin, Italy; simone.soccal@unito.it (S.S.); francesca.bertolini@unito.it (F.B.); vitina.carriero@unito.it (V.C.); stefano.levra@unito.it (S.L.); fabioluigimassimo.ricciardolo@unito.it (F.L.M.R.); 3Severe Asthma, Rare Lung Disease and Respiratory Pathophysiology, San Luigi Gonzaga University Hospital, Orbassano, 10043 Turin, Italy; pizzimentistefano@gmail.com; 4Institute of Translational Pharmacology, National Research Council (IFT-CNR), Section of Palermo, 90146 Palermo, Italy

**Keywords:** nasal polyps, chronic rhinosinusitis, asthma, type-2 inflammation, biologic treatment, dupilumab, endoscopic sinus surgery, biomarkers

## Abstract

**Background:** Chronic rhinosinusitis with nasal polyps (CRSwNP) and asthma are mutually correlated with Type-2 inflammation. Dupilumab is effective in uncontrolled and relapsing CRSwNP. However, the precise characterization of Type-2 inflammation and the impact of previous surgery on clinical outcomes need clarification. **Methods**: We present a prospective observational study on a 38 CRSwNP-patient cohort, whose Type-2 endotype was confirmed after a multidisciplinary approach shared among ENTs, pneumologists and allergologists. Patients were treated with dupilumab and evaluated at 15 days and 1-3-6-12-18-24-30 months, focusing on clinical (VAS, nasal polyp score—NPS), radiological (Lund-Mackay) and quality of life (SNOT-22) parameters, as well olfactory function, asthma control, variation of Type-2 markers and number and extent (ACCESS score) of previous surgeries. **Results**: We confirmed the efficacy of dupilumab in total and sub-items VAS, NPS, SNOT-22 and sniffing score, as well as Lund–Mackay score improvements, observable and significant after 2 weeks of treatment (*p* < 0.0001) and long-lasting over 30 months. Good to excellent response criteria to biologic treatment at 6 months was observed in 30/32 patients. Comorbid asthma reached rapid control (*p* < 0.0001) and exhaled nitric oxide normalization was achieved. One single “not adequate” surgery showed a trend to milder improvement, as well as a higher ACCESS score to better olfactory outcome. **Conclusions**: The accurate selection of uncontrolled relapsing CRSwNP in terms of Type-2 endotyping by multidisciplinary approach can maximize dupilumab efficacy. The number and extent of previous surgeries may differentiate the response, although this effect is difficult to catch in real life. “Adequate” ESS surgery before dupilumab may drive mostly effective disease control.

## 1. Introduction

Chronic rhinosinusitis with nasal polyps (CRSwNP) strongly affects patients’ quality of life, especially in uncontrolled cases [1]. Type-2 inflammation plays a crucial role in CRSwNP pathogenesis and represents the most common endotype characterized by high levels of specific interleukins (IL-4, IL-13, IL-5), high tissue and blood eosinophilia (B-EOS) and high blood levels of IgE and of the exhaled fraction of nitric oxide (FENO) [2].

The efficacy of dupilumab, a monoclonal antibody anti IL-4 receptor alpha chain (IL-4Rα) blocking both IL-4 and IL-13 signalling, in difficult-to-treat and relapsing CRSwNP has been demonstrated in randomized clinical trials (RCTs), which showed a significant improvement in clinical outcomes after a few weeks of treatment [3]. Large-scale real-life studies confirmed and extended the results in terms of endoscopic polyp score, symptoms, quality of life (QoL) and olfactory function during up to 12 months of observation [4].

Different evidence [5,6] states a profound correlation between upper and lower respiratory inflammatory tract diseases and, in this context, is fundamentally a clinical multidisciplinary approach. According to the literature, up to 67% of CRSwNP patients have comorbid asthma with a higher risk of recurrence of nasal polyps after surgery [7]. Dupilumab is a biologic treatment for uncontrolled severe type-2 asthma able to reduce significantly the rate of asthma exacerbations and the burden of oral corticosteroid (OCS) use [8]. Greater efficacy is observed in patients with elevated type-2 inflammatory biomarkers (B-EOS and FENO). In addition, in patients with CRSwNP and coexisting asthma, dupilumab significantly improved both asthma and NPs outcomes [9].

EUFOREA [10] inclusion criteria for treatments of CRSwNP with biologics include both the evidence of type-2 biomarkers and asthma comorbidity. Albeit desirable, the histologic finding of NP tissue eosinophilia is not routinely performed before starting a biologic. Actually, high tissue eosinophilia in NPs, although associated with a more severe disease and a higher recurrence rate [11], is considered a predictor of good response to biologics [12] compared with a low eosinophilic phenotype.

Our study aimed to evaluate the effectiveness of dupilumab in CRSwNP after a multidisciplinary approach to asthma assessment and confirmation at baseline of T2 inflammation in nasal polyp tissue. We conducted a prospective study of up to 3 years of treatment in a real-life setting, focusing on improvement of symptoms and QoL (Sinonasal Outcome Test 22—SNOT-22, visual analogue scale—VAS, nasal congestion score—NCS), endoscopic (nasal polyp score—NPS) and radiological features (Lund–Mackay score—LMS), as well as olfactory function (Sniffin test—SSIT-16), asthma control (ACT) and variation of T2 biomarkers (B-EOS, total IgE and FENO). The effect of previous surgeries on dupilumab efficacy was also evaluated.

## 2. Materials and Methods

### 2.1. Patients and Study Design

A prospective observational study was conducted on a cohort of 38 consecutive adult patients with a diagnosis of severe uncontrolled chronic rhinosinusitis with nasal polyps (CRSwNP), according to EPOS [1], and treated with dupilumab between February 2021 and December 2023 at the Unit of Otorhinolaryngology—Head and Neck Surgery at the A.O.U. San Luigi Gonzaga Hospital, Orbassano (Turin), Italy.

Patients were enrolled with the following characteristics: age > 18 years, endoscopic diagnosis of CRSwNP by NPS > 5 or a SNOT-22 score > 50, failure of prior medical treatments due to complications or inefficacy (at least 2 cycles of systemic corticosteroid in the last year) and failure of previous surgical treatment [4]. A previous biologic treatment had to be interrupted after an adequate washout period before enrolment. In addition, the type-2 endotype was confirmed in all patients according to EPOS/EUFOREA criteriaby absolute blood eosinophil count (B-EOS) > 0.150 × 10^9^/L and tissue eosinophils > 10/HPF, when available on histological samples from intraoperative samples, or from outpatient pretreatment biopsy. The exclusion criteria were pregnancy, active autoimmune/rheumatic disease, active oncologic disease in the last 5 years, immunosuppressive therapy and concomitant systemic corticosteroid therapy for chronic autoimmune disorders.

The Institutional Review Board (IRB) of San Luigi Gonzaga University Hospital accepted the study protocol (protocol number 2690/18/02/2021). Written informed consent was obtained from each participant before they were included in the study, following the principles of the Declaration of Helsinki.

### 2.2. Clinical, Endoscopic, Radiologic and Bio-Humoral Evaluation During Treatment

After baseline evaluation (V0), the following visits were conducted at 2 weeks (V1), 1 month (V2), 3 (V3), 6 (V4), 9 (V5), 12 (V6), 18 months (V7),2 years (V8), 30 months (V9) and 3 years (V10), if available. Patients who discontinued treatment before 6 months were excluded from the study. Dupilumab was administered through a subcutaneous injection with a starting dose of 600 mg followed by 300 mg once every 2 weeks. The first injection was carried out under medical control in the outpatient clinic. Adverse events were monitored and defined as early or late if the onset was before or after 30 days of treatment, respectively.

Before treatment, all patients were investigated in terms of demographic, anthropometric data, clinical history and habits, previous endoscopic sinus surgery (ESS), use of nasal or systemic corticosteroids and previous biologics and the presence of allergy to inhalant agents or intolerance to non-steroidal anti-inflammatory drugs (NSAIDs). At each visit from V1 to V10, clinical scores including the total and single domain nasal symptoms visual analogue scale (VAS), SNOT-22 (total and QoL subtitle) and nasal congestion score (NCS) were collected. VAS measures the intensity of symptoms on a horizontal 10 cm line. Eight domains for each nasal symptom were measured, as previously reported [13]: nasal obstruction, rhinorrhoea, sleep disturbances, olfaction, facial pain, sneezing, headache and post-nasal drip. The VAS for the total nasal symptom score was calculated by the sum of each single domain (range 0–80). A total VAS score < 20 (on a range of 80) and SNOT-22 < 20 could be considered as reference for good CRSwNP control in real-life studies [13]. The Nasal Congestion Score (NCS) referred to patient-assessed symptom severity score for nasal congestion (NCS score 0–3; 0 = no symptoms, 1 = mild, 2 = moderate and 3 = severe). A Sniffin text (SSIT-16) was performed as previously described. A nasal rigid endoscopy with a 0–45° endoscope (Karl Storz° Endoskope, Tuttlingen, Germany) was performed at every visit to evaluate both nasal cavities and to measure the NPS on a scale from 0 to 8 [3]. A maxillo-facial computed tomography (CT) scan was performed at baseline in order to evaluate the radiological severity of the disease (Lund–Mackay score—LMS) [14] and the completeness of previous endoscopic surgeries, when performed, based on the ACCESS score, a radiologic quantitative score of paranasal sinuses patency [15]. An ACCESS score ≤ 10 is indicative of prior adequate sinus surgery. The CT scan was performed again after 6 months of therapy, then at 1 year and 2 years. The Inflammatory phenotype was evaluated at each visit by an absolute blood eosinophils automated count (B-EOS, ×10^9^/L), total serum IgE level (by Elisa, KU/L) and FENO (ppb) assessment, when available. FENO was measured with the single breath technique using FENO+ (Medisoft, Sorinnes, Belgium).

All the patients were evaluated in a multidisciplinary approach by a pneumologist and an allergologist in order to confirm or exclude a concomitant diagnosis of asthma [16] and atopy. Atopy for seasonal or perennial allergens was demonstrated by diagnostic tests (Prick or Specific IgE). When asthma was confirmed, both an asthma control test (ACT) and a respiratory function test (absolute and relative) were performed at baseline and during follow-up. A lung function test was assessed in patients with CRSwNP and concomitant asthma by performing spirometry (Vmax Encore 62, Carefusion, Wurzburg, Germany). The absolute FEV1, FEV1%pred, absolute FVC, FVC% pred, FEV1/FVC ratio (IT) absolute and %pred ratio were collected at baseline and at least one time after 12 months from the dupilumab initiation.

The response to biologic treatment was evaluated at 6 months (V4) according to EPOS 2023 criteria [17].

### 2.3. Statistical Methods

The results are reported as mean with standard deviation for continuous variables or frequencies with a percentage for categorical variables. Comparisons between continuous variables were performed primarily using the independent-sample or paired-sample *t*-test or the Mann–Whitney test (or, alternatively, the Kolmogorov–Smirnov test in the case of groups with very small populations), depending on the distribution. Univariate correlations between continuous parameters were examined using Pearson’s or Spearman’s correlation tests, depending on the distribution. For categorical variables, Fisher’s exact test or the chi-square test was used, depending on the sample size. The ANOVA or Krustal–Wallis test was used for comparisons between multiple groups, depending on the normality of the distribution, and Tukey’s Multiple Comparison Test or Dunn’s Multiple Comparison Test was applied for comparison between single groups. Python Version 3.8. was used for the paired sample test to evaluate the variation of Delta parameters for each time point of treatment. The results were considered statistically significant for *p* < 0.05. The analysis was performed using Graph Pad Prism software (version 9.0; GraphPad Software Inc., San Diego, CA, USA) and SPSS Statistic Version 28 (IBM Corp., Armonk, NY, USA). Evaluation at 3 years (V10) was reached by one single patient. This patient was, therefore, excluded from the statistics.

## 3. Results

### 3.1. Patients’ General Characteristics

A total of 38 patients were enrolled in the study; 6 patients did not reach a sufficient follow-up period (6 months) to be included in the study. Therefore, 36 patients were included in the study (Table 1) (Female 10, Males 22). The mean age was 57.9 years old, with a mean BMI of 25 ± 2.5.

Atopy was present in 62.5%, and 31.3% had NSAIDS intolerance. Overall, five patients suffered from allergic rhino-conjunctivitis and one patient from atopic dermatitis.

A total of 71.9% (N= 23) had received oral corticosteroid courses (mean 2.4 ± 2.5) for uncontrolled symptoms in the last year, reaching a mean cumulative dose of 237.0 mg ± 405.7 mg. In addition, 84.4% (N = 27) were treated with nasal topical steroids at baseline. Out of 32 patients, 28 had previous ESS, with a mean of 2.6 ± 2.3. The mean time from the last surgery was 76.0 ± 80.9 months. The patients were grouped according to the number of surgeries.

Asthma was reported in 25 patients (78.1%), with a mean ACT of 13.9 ± 5.1 before treatment. Data about the respiratory function of asthmatic patients and their treatment are reported in Table 2. Six patients were treated previously with biologic drugs for severe asthma.

### 3.2. Symptoms, Endoscopic, Radiologic Scores and Biomarkers of Nasal Polyps at Baseline

Before biologic treatment, the mean total nasal symptoms, as expressed as a total VAS score, was 41.6 ± 14.4, encompassing 7.3 ± 2.5 for nasal obstruction, 5.7 ± 2.6 for rhinorrhoea, 8.1 ± 2.6 for olfaction and 5.8 ± 2.8 for sleep symptoms. The mean basal SNOT-22 value was 55.9 ± 15.1, and the sniffing stick value yielded a mean of 3.9 ± 4.0. At endoscopy, the mean NPS score was 5.3 ± 1.5. Finally, CT scan evaluation revealed a mean LMS score of 16.7 ± 5.1, while the ACCESS score, among 25/28 patients who underwent FESS, yielded a mean of 9.8 ± 6.7 (Table 3).

The mean absolute blood eosinophil value at baseline was 0.550 ± 0.330 × 10^9^/L, while the mean total IgE was 167.43 ± 235.0 KU/L. Of the 13 asthmatic patients in which data were available, the mean FENO at baseline was 50 ± 25.7 ppb. According to the recognized cut-off of positive T2 biomarkers, 26/32 patients had absolute B-EOS ≥ 0.150 × 10^9^/L, 15/27 patients had total IgE > 100 KU/L and 11/13 had FENO > 30 ppb.

Univariate correlations between continuous parameters showed a significant association between total VAS and SNOT-22 (*p* = 0.004). The correlation between the B-EOS and total IgE (*p* = 0.059) and the NPS with Sniffin (0.058) score, respectively, reached the limits of statistical significance (Table 4).

### 3.3. Clinical Response by Symptoms, Endoscopic, Radiologic Scores and Biomarkers Analysis During Follow-Up

A significant difference from V0 to V2 for paired samples was found for the total VAS (41.6 vs. 18.4, *p* < 0.0001), olfactory VAS (8.00 vs. 5.07, *p* < 0.0001), nasal obstruction VAS (7.31 vs. 3.86, *p* < 0.0001), rhinorrhoea VAS (5.86 vs. 3.69, *p* = 0.004), sleep VAS (5.76 vs. 2.72, *p* < 0.0001), facial pain VAS (4.97 vs. 1.79, *p* < 0.01) and from V0 to V1 for SNOT-22 (55.6 vs. 32.9 *p* < 0.0001), NPS (5.2 vs. 3.9, *p* < 0.0001) and the Sniffin’ Sticks value score (3.9 vs. 7.5, *p* = 0.0001). All the above scores maintained a significant improvement compared to V0 at all the following time points from V2 to V9 (Table 3 and Figure 1A–D).

When applying the EPOS 2023 response criteria to biologic treatment at 6 months (V4) [16], 30 out of 32 patients reported a good-to-excellent response.

LMS showed a significant improvement from V0 to V4 (6 months) (15.4 vs. 7.4 *p* < 0.0001) maintained at further evaluation (V6 and V8, respectively) (Table 3 and Figure 1E).

The blood eosinophils did not show any significant change during the follow-up, as a whole patient group (*p* > 0.05 from V0 to V2–V9), although 16 out of 32 patients (50%) experienced a transient significant increase (at least 50% increase from basal) at different time points from V1 to V5 (*p* < 0.001), never exceeding an absolute count of 2.50 × 10^9^/L (Figure 2A,B). The total IgE measured showed a progressive decrease from V3 to V9, statistically significant at V4, V5 and V6 (*p* < 0.05), as shown in Table 3 and Figure 2C

### 3.4. Outcomes According to ACCESS Score Evaluation

Patients who underwent FESS before treatment were categorized according to ACCESS scores ≤ 10 (n = 14) or >10 (N = 11). The basal characteristics of the patients, divided into two groups, are reported in Table 5. No significant difference was reported between groups except a longer mean time between the last surgery and biologics (*p* = 0.026) for patients with higher ACCESS scores and a non-significant trend for a lower course number of OCS.

Nasal outcomes according to ACCESS score did not reveal any significant difference between the two groups, as reported by the comparison of paired delta change of the total VAS, SNOT-22, NPS and Sniffin scores (Table 6). However, although not significant, patients with higher basal ACCESS scores showed better improvement at V2 and V4 in their Sniffin’s Sticks test scores compared to those with lower ACCESS scores (2.8 ± 5.0 vs. 6.6 ± 3.7, *p* = 0.0658 at V2 and 3.2 ± 3.9 vs. 6.8 ± 4.3, *p* = 0.0736 at V4).

Three representative cases of radiologic and endoscopic outcomes before and after dupilumab treatment, according to basal Access Score and previous surgery, are reported in Figure 3A–C. Out of 32 patients, 2 underwent “adequate” Full House FESS of revision for relapsing CRSwNP during dupilumab treatment (one reported in Figure 3B).

### 3.5. Outcomes According to Previous Surgeries Stratification

Patients who underwent ESS before treatment were categorized according to the number of previous surgeries and compared in terms of paired delta change of total VAS, SNOT-22, NPS and Sniffin’s Sticks test scores (Table 7). No significant difference was reported between the groups. However, patients who underwent one surgery before dupilumab experienced a lower initial improvement in terms of VAS (*p* =0.0795 at V2 and *p* = 0.0719 at V4) compared to the other groups. A similar trend was observed for SNOT-22 at V1, V3 and V4 and NPS at V2. On the other hand, patients with the highest numbers of surgeries experienced milder general increases in the Sniffin’s Sticks test, although not significant.

### 3.6. Outcomes According to Asthma Diagnosis

Patients with concomitant asthma and CRSwNP (N = 25) were further analysed. We found a significant reduction for total IgE (167 vs. 47 KU/L, *p*-value = 0.017) and FENO (50 ppb vs. 27.25 ppb, *p*-value = 0.01) during the follow-up (second measure at least 6 months after basal) (Figure 4).

Although a positive trend in lung volumes and IT could be observed, we did not find a significative variation in our total cohort of patients regarding FEV1 (2.72 vs. 3.01, *p* value = 0.25), FVC (3.83 vs. 4.09, *p* value = 0.41) and IT (70.19 vs. 72.97, *p* value = 0.21).

## 4. Discussion

In line with the results of previous RCTs [3] and observational studies [4], dupilumab administered every 2 weeks significantly improves symptoms and health-related quality of life, as reflected by an improvement in questionnaire answers (VAS, SNOT-22) and olfactory perception (SSIT-16, VAS olfaction). In addition, biologic treatment strongly impacts clinical outcomes, as determined by a reduction of polyps’ size. These improvements in our cohort were visible and significant after 2 weeks of treatment, and, after a rapid decrease in the first months, they keep stable with time up to more than 2 years [18].

According to De Corso et al. [4] and, similarly, to RCTs studies [3], all patients benefited from dupilumab, irrespective of previous surgery, but a faster NPS and SNOT-22 decrease was seen in patients who had undergone surgery, while a faster decrease in VAS olfaction was seen in patients without previous surgery, as well as a faster improvement in SSIT-16 odour discrimination. The authors also affirm that these differences, whilst statistically significant, are not relevant from a clinical point of view, and the number of patients without previous surgery is small, which is also a limitation of our study. In our cohort of patients, a trend for a faster VAS and SNOT-22 decrease and improvement in SSIT-16 odour discrimination can be appreciated in patients without previous surgery.

Updated indications for Biologics in Chronic Rhinosinusitis with Nasal Polyps by EPOS/EUFOREA [17] state a detailed definition of type-2 inflammation in NPs by the cut-off expression of type-2 biomarkers. Tissue eos ≥ 10/hpf or absolute B-EOS ≥ 0.150 × 10^9^/L or total IgE > 100 KU/L are reported as a good surrogate for the type-2 endotype. RCTs leading to dupilumab approval in CRSwNP did not include type-2 biomarkers among inclusion criteria [3], but a post hoc analysis of SINUS-52 that used the JESREC algorithm to classify patients with or without eosinophilic chronic rhinosinusitis (ECRS) concluded that there is a fair benefit [19]. In real-life, observational, multicentre studies [4], the mean absolute eosinophils (0.500 × 10^9^/L) and IgE (180 KU/L) of recruited patients exceeded the type-2 cut-off, but it was not a criterion for inclusion.

In clinical practice, serologic biomarkers are often preferred for their easy feasibility, but their ability to predict dupilumab NP response is debated, with elevated baseline B-EOS associated with a trend of better SNOT-22 improvement and total serum IgE moderately associated with almost all clinical variables good outcome after 1 year with almost all clinical variables [20]. At baseline, we did not report any significant correlation between B-EOS and total IgE compared to symptoms and endoscopic scores. Although some studies reported an association between symptoms and increased B-EOS, the major role for this biomarker was in distinguishing CRSwNP phenotypes, as well as predicting polyp recurrence [21]. The search for local biomarkers that can better characterize the spectrum of severity of CRSwNP compared to B-EOS is still ongoing, and a few have been suggested (i.e., Calprotectin and IL-5); therefore, the negative correlation (r), although not significant, among B-EOS and symptoms [22] is not surprising.

In this prospective study, we enrolled patients according to absolute B-EOS > 0.150 × 10^9^/L and tissue eosinophils > 10/HPF, when available on histological samples from intraoperative samples or by outpatient pretreatment biopsy, thus reinforcing the pretreatment selection for a type-2 target therapy. Total IgE are generally considered to have low sensitivity in predicting type-2 inflammation [23], as demonstrated in huge asthma cohorts, and their role in CRSwNP is more related to the interaction with specific triggers (Aspirin, moulds, *Staphilococcus aureus* endotoxins) [24] that trigger the type-2 pathway.

Among the 32 patients who reached at least 6 months of follow-up, only 2 could be considered moderate responders according to EPOS 2023 criteria [17]; all the other patients had a good-to-excellent response (at least four response criteria). Compared to other studies, the extent of this improvement is higher [25]. We believe that this better response is due to careful patient selection in terms of T2 inflammation biomarkers. Accordingly, decreased eosinophilia and thickened basement membranes are found in histopathologic analysis of biologic non-responders [12]. These results may push clinicians towards a more precise process of NP endotyping before biologic treatment. In addition, we reported a sustained response over time that could configure a picture of clinical remission of the disease, even during treatment.

All patients underwent a multidisciplinary evaluation for concomitant asthma. Although the majority of patients included in the study did not meet the criteria for severe asthma [12], needing a previous biologic treatment (n = 6), asthma control improved in all patients, paralleled by a reduction in FENO values. The importance of an accurate asthma assessment in CRSwNP lies not only in being one of the criteria for biologic treatment but also in the knowledge that patients with NPs benefit from concomitant appropriate asthma treatment.

The issue of whether to undergo extensive surgery of the paranasal sinuses before undertaking biological therapy is a matter of debate in the rhinological community. Many recommendations advocate for complete surgery before candidacy for biologic drugs. EPOS 2020 suggested the candidacy for biologic drugs of patients affected by type-2 CRSwNP after the failure of surgical treatment. On the other hand, EUFOREA in 2019 [10] extended its indication to naïve patients with type-2 CRSwNP when four out of five severity criteria were met. This may explain why the number of naïve patients is still low compared to the operated ones, so this cohort is still hard to evaluate. Some studies, such as the one by Migliani et al. [26], demonstrated the superiority of ESS in SNOT-22 and NPS compared to biologics. This might incentivize clinicians to perform surgery as a first-line treatment if medical therapy is not sufficient, in consideration of the high cost of biologics [27] and the duration of therapy, which can be long-lasting and is still undefined.

According to the number of previous surgeries, the best outcomes in terms of SNOT-22 were observed (although without being statistically significant) in patients with two or more previous surgeries compared to those with one, at least in terms of VAS and SNOT-22. A suggestive hypothesis is that, in this cohort of patients, the first surgery may not be “adequate” compared to a more complete and extensive second intervention. On the other hand, the milder improvement in the Sniffin test, for patients with three or more surgeries, may be linked to more fibrotic residual tissue with hyperostosis and osteitis, which is less responsive to biologic treatments [28].

The extent of surgery is a topic still debated: EPOS2020 guidelines require at least one complete surgery (FESS/ESS), but it depends on clinicians’ experience, and, especially in the past, relapses were treated with anterior FESS (limited to the ethmoid and maxillary sinus) or polypectomy, so it was common to report patients who underwent too many surgeries (>3) before starting dupilumab. We suppose, in the future, patients will start biologics before undergoing multiple uncompleted surgeries, due to the improvement of surgical expertise as well as to the acquisition of new knowledge about the effectiveness of biologics.

In our experience, no statistical correlation was found between ACCESS score and clinical response, contrary to a study by Gian Marco Pace et al. [29], where ACCESS and response to therapy were correlated in patients treated for at least 12 months. In our experience, even in the presence of low ACCESS scores (<10), the sinuses most frequently not fully opened were sphenoid and frontal, requiring more expertise by surgeons, higher operative time and longer and more difficult postoperative management. This aspect may impact the persistence of inflammation and congestion throughout treatment. We can suggest that more adequate surgery, through one-to-one opening and enlarging of all sinus ostium, can contribute to less persistent eosinophilic inflammation [30] with an additional effect to that of biologics.

Another study by Alicandri-Ciufelli [31] reported a statistically significant correlation between a low ACCESS score and better NPS and SNOT-22 measured after 1 month, while the VAS loss of smell measured after 1 year was worse in this subgroup. In line with this finding, we observed a better improvement at V2 and V4 in Sniffin’s Sticks tests for those with higher access scores.

A future challenge may be to “adapt” surgical extension to the characteristics of the patient and the sinus involvement, but further research is required in order to understand the real correlation between surgical extension and clinical outcomes.

## 5. Conclusions

Asthma and CRSwNP share the same T2 inflammatory pathway, underlining the relevant impact of dupilumab on clinical outcomes and quality of life in patients affected by both diseases, as well as on biological markers and radiological features. An accurate endotyping by eosinophilic analysis of nasal (biopsy, cytology) or bronchial samples (induced sputum) may increase the predictive yield of T2-driven biologics.

The number and extent of previous surgeries (ACCESS) score may play a role in influencing the response to therapy, even if, from a strictly clinical point of view, a satisfactory response is seen in all cohorts of NPs patients. Further research is needed to clarify these correlations, but we assume that, in future, there will be fewer patients with a history of multiple interventions and therefore with a higher risk of nervous fibre damage. We suggest that performing at least one “adequate” Full House FESS surgery before starting dupilumab represents the way for a better control of the disease and better sinus ventilation.

## Figures and Tables

**Figure 1 biomedicines-13-00501-f001:**
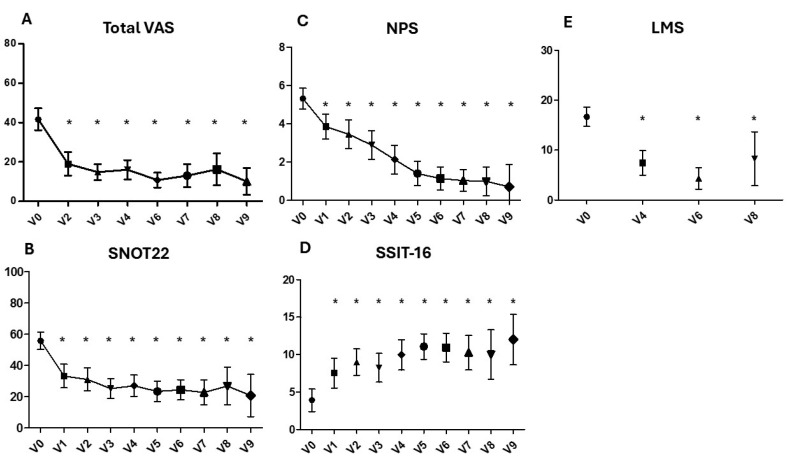
Clinical, endoscopic and radiologic scores evaluated before and after dupilumab treatment. From V1 to V9. Total nasal symptoms visual analogue scale (total VAS) (**A**), SNOT-22 (**B**), nasal polyps score (NPS) (**C**); Sniffin’s Sticks test (SSIT-16); (**D**), Lund–Mackay score (LMS) (**E**). * *p* < 0.0001 (paired-sample *t*-test).

**Figure 2 biomedicines-13-00501-f002:**
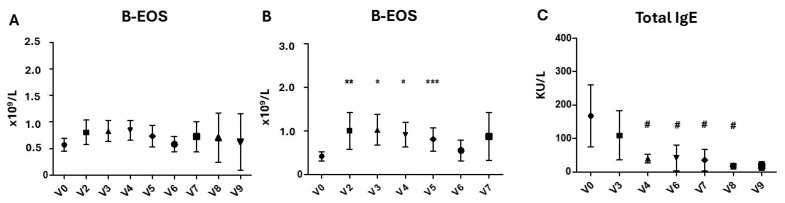
Absolute blood eosinophil count (B-EOS, ×10^9^/L) in total CRSwNP population (**A**), a patient subgroup according to at least 50% increase from basal (**B**) and Total IgE (KU/L) (**C**) evaluated before and after dupilumab treatment. From V1 to V9. * *p* < 0.0001; ** *p* < 0.001; *** *p* < 0.01, # *p* < 0.05 (paired-sample *t*-test).

**Figure 3 biomedicines-13-00501-f003:**
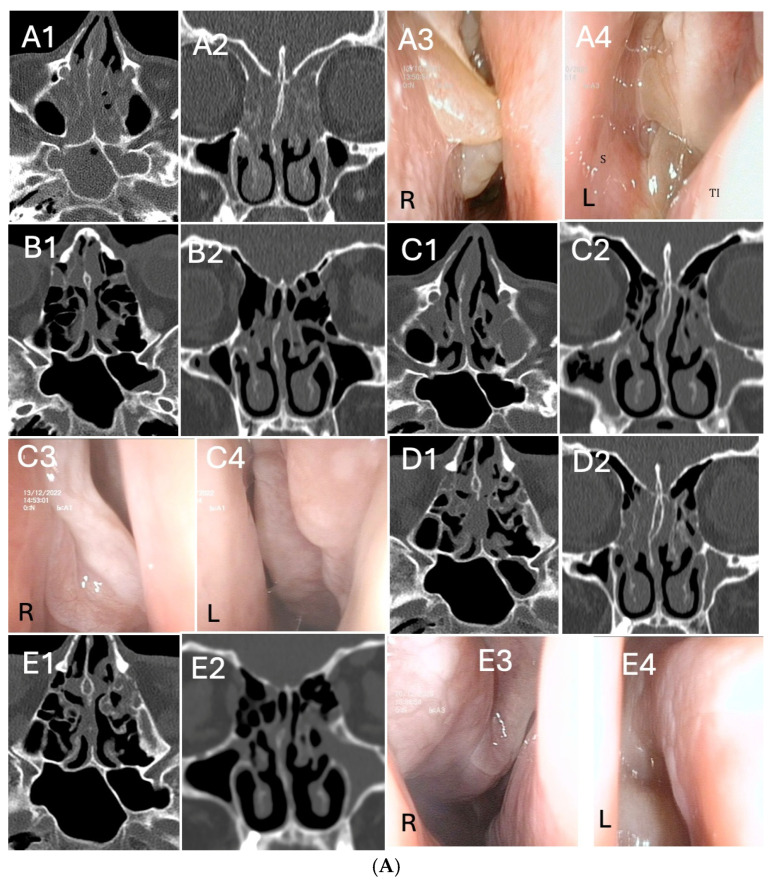
(**A**). Radiological and endoscopic (R = right, L = left) images in relation to dupilumab therapy without surgery in CRSwNP (eosinophils>30/hpf). (**A1**–**A4**): start of dupilumab. Axial (**A1**) and coronal (**A2**) computed tomography (CT) images show pansinusitis. Endoscopic examination (**A3**,**A4**) of both nasal cavities shows polypoid formation completely occupying the nasal fossa (total NPS = 7) S:sinus; TI:turbinate. (**B1**,**B2**): 6 months after initiation of dupilumab. Sphenoidal (**B1**), frontal and maxillary sinuses (**B2**) completely disease-free. Pictures (**C1**–**C4**): 12 months after initiation of dupilumab. Radiologic (**C1**,**C2**) and endoscopic examination (**C3**,**C4**) of both nasal cavities show significant reduction of polypoid mass. (**D1**,**D2**): axial (**D1**) and coronal (**D2**) computed tomography (CT) images 18 months after initiation of dupilumab. Outcomes are maintained in the long run. (**E1**–**E4**): 24 months after initiation of dupilumab. Maintenance of the radiological (**E1**,**E2**) and endoscopic (**E3**,**E4**) outcomes is evident, even at long term follow-up visit. (**B**). Radiological and endoscopic (R = right, L =left) images in relation to dupilumab therapy after “adequate” Full House FESS of revision for CRSwNP. (**A1**–**A4**): CT (**A1**,**A2**) and MRI (**A3**,**A4**) images showing CRSwNP recurrence after two “not adequate” FESS (ACCESS score 20/24). Intra-operatory images (RESS): (**A5**) right nasal fossa NPS = 4; (**A6**) debridement of nasal polyps; (**A7**) left maxillary sinus empyema; (**A8**) frontal sinusotomy. (**B1**–**B4**): start of dupilumab 18 months after “adequate” Full House FESS, following CRSwNP relapse (eosinophils>50/hpf). Endoscopic (**B1**,**B2**) and radiological (**B3**,**B4**) images, ACCESS score 2/24. (**C**): 6 months after initiation of dupilumab. Radiologic (**C1,C2**) and endoscopic examination (**C3**,**C4**) of both nasal cavities shows partial reduction of polypoid mass. (**D1**–**D4**): 12 months after initiation of dupilumab, a local control of the disease is evidenced in both axial (**D1**) and coronal (**D2**) computed tomography (CT) and endoscopic (**D3**,**D4**) images (**E1**–**E4**): 24 months after initiation of dupilumab, images highlight the maintenance of the radiological (**E1**,**E2**) and endoscopic (**E3**,**E4**) results even at long term follow-up visits. Inferior turbinates are preserved, while middle turbinate was remodeled for correct sinus ventilation and disease control. “ss”: sphenoid sinus; “sm”: maxillary sinus. (**C**). Radiological and endoscopic (R = right, L = left) images in relation to dupilumab therapy after “adequate” salvage Full House FESS in patient who underwent two previous “not adequate” surgeries (last surgery performed 18 months before starting dupilumab). (**A1**–**A4**): The patient started biologic treatment with dupilumab. Coronal (**A1**) and axial (**A2**) CT scans evidenced ethmoidal cells still not opened and hyperostosis of the anterior and posterior compartment. Endoscopic images (**A3**,**A4**): NP relapse and tissue remodeling. (**B1**–**B4**): 6 months after initiation of dupilumab, persistence of nasal polyps seen radiologically (**B1**,**B2**) and endoscopically (**B3**,**B4**). (**C1**–**C4**): 18 months after initiation of dupilumab, ACCESS score 22/24 (**C1**,**C2**). Endoscopic sinus surgery (**C3**,**C4**) was performed in this patient, keeping on treatment with dupilumab. (**D1**–**D4**): 1 year after “adequate” Full House FESS and 3 years after dupilumab. TC scans (**D1**,**D2**) evidenced paranasal sinuses fully opened. Corresponding endoscopic image (**D3**,**D4**).

**Figure 4 biomedicines-13-00501-f004:**
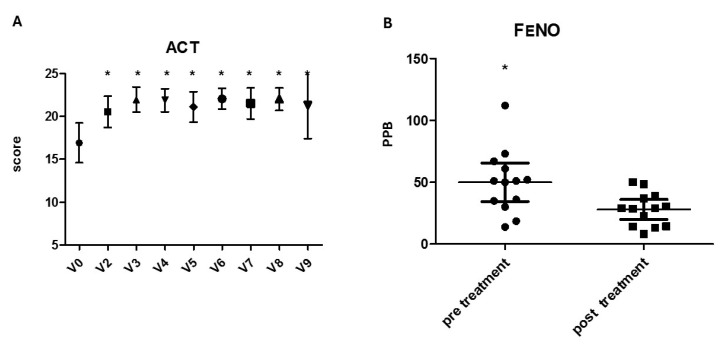
Asthma control test (ACT, score) (**A**) and exhaled nitric oxide (FENO, parts per billion, PPB) (**B**) evaluated in patient with concomitant asthma and CRSwNP before and after dupilumab treatment from V0 to V9 and before and after treatment (at least 6 months of follow-up); * *p* < 0.0001 (paired-sample *t*-test).

**Table 1 biomedicines-13-00501-t001:** General characteristics at baseline of CRSwNP patients.

	N = 32
**Age** **Age at diagnosis**	57.9 [38–78] 39.8 [4–65]
**Sex**	Males 22 (68.7%) Females 10 (32.3%)
**BMI**	25.0 ± 2.5
**Smokers**	2 (6.3%)
**Familial Atopy**	13 (40.6%)
**Atopy**	20 (62.5%)
**Allergic comorbidities**	6 (18.7%) Rhino-conjunctivitis 5 Atopic dermatitis 1
**NSAIDs intolerance**	10 (31.3%)
**AERD**	9 (28.1%)
**Presence of asthma**	25 (78.1%)
**B-EOS (×10^9^/L)**	0.552 ± 0.333
**Total IgE (KU/L) ***	167.4 ± 235.0
**INCS use**	27 (84.4%)
**OCS needed in the last year**	23 (71.9%)
**Prednisone in the last year (mg) ****	249.3 ± 417.8
**OCS courses in the last year *****	2.4 ± 2.5
**days of OCS in the last year *****	40.2 ± 81.1
**Previous surgeries**	28 (87.5%) No surgery 4 (12.5%) 1 surgery 8 (25%) 2 surgeries 7 (21.9%) 3 or more surgeries 13 (40.6%)
**Numberof surgeries *****	2.6 ± 2.3
**Time between last surgery and biologics (months)**	76.0 [0–300]

Results of non-categorical variables are reported ad mean ± standard deviation. Categorical variables are reported as number (N) and percentage. Range is reported in []. NSAIDs: non-steroidal anti-inflammatory drugs; AERD: Aspirin-exacerbated respiratory disease; B-EOS: absolute blood eosinophil count; INCS: intranasal corticosteroids. Data available in * N = 27; ** N = 28; *** N = 29. Years and months are reported as mean and range [min, max].

**Table 2 biomedicines-13-00501-t002:** Characteristics at baseline of CRSwNP patients with associated asthma.

	N = 25
**Age**	57.5 [38–78]
**Age at asthma onset**	42.5 [22–62]
**Asthma control (ACT)**	16.9 ± 4.5
**Previous Biologic therapies**	6 (18%) Omalizumab n = 6 Benralizumab n = 3
**Use of inhalers**	23/25 (92%) SABA 3 (12%) ICS 3 (12%) ICS + SABA 8 (32%) ICS + LABA 9 (36%)
**FVC (L) ***	3.70 ± 0.83
**FEV1 (L) ***	2.70 ± 0.77
**FEV1/FVC ***	76.9 ± 7.9
**FENO (ppb) ****	50.0 ± 25.7
**Total IgE (KU/L) *****	186.8 ± 266.8
**B-EOS (×10^9^/L)**	0.570 ± 0.036

Results of non-categorical variables are reported ad mean ± standard deviation. Categorical variables are reported as number (N) and percentage. Range is reported in []. SABA: short-acting bronchodilators agonists; LABA: long-acting bronchodilators agonists; ICS: inhaled corticosteroids; FVC: forced vital capacity; FEV1: forced respiratory volume in the first second; FENO: exhaled nitric oxide; B-EOS: absolute blood eosinophil count. Data available in * N = 18; ** N = 13; *** N = 20. Years and months are reported as mean and range.

**Table 3 biomedicines-13-00501-t003:** Change in clinical, serologic and radiologic parameters at baseline and over time during treatment in the study population (N = 32).

	V0	V1	V2	V3	V4	V5	V6	V7	V8	V9	*p* *
**SNOT-22**	55.9 ± 15.1	32.9 ± 20.2	31.1 ± 18.8	25.2 ± 16.7	37.1 ± 17.9	23.4 ± 16.6	24.5 ± 16.5	22.7 ± 18.8	26.8 ± 19.8	20.8 ± 14.6	<0.001
**NPS**	5.3 ± 1.5	3.9 ± 1.7	3.5 ± 1.9	2.9 ± 2.0	2.1 ± 1.9	1.4 ± 1.6	1.1 ± 1.5	1.1 ± 1.4	1.0 ± 1.2	0.6 ± 1.0	<0.001
**NCS**	2.3 ± 0.7	1.4 ± 0.8	1.2 ± 0.8	1.0 ± 0.7	0.7 ± 0.7	0.7 ± 0.8	0.6 ± 0.7	0.4 ± 0.6	0.5 ± 0.5	0.3 ± 0.5	<0.001
**SSIT-16**	3.9 ± 4.1	7.5 ± 5.2	9.0 ± 4.5	8.2 ± 5.1	9.9 ± 5.0	10.9 ± 4.4	11 ± 4.8	10.5 ± 5.1	10.8 ± 4.7	10.7 ± 5.0	<0.001
**INCS adherence**	80%	79%	93%	83%	80%	96%	100%	76%	89%	87%	NS
**VAS total**	41.6 ± 14.4	25.0 ± 15.9	19.0 ± 15.2	14.9 ± 10.4	16 ± 12.4	-	10.7 ± 9.2	11.9 ± 13.0	14.2 ± 11.4	12.3 ± 9.5	<0.001
**Smell** **(VAS)**	8.1 ± 2.6	5.0 ± 3.7	3.6 ± 3.5	2.8 ± 3.4	3.1 ± 3.5	2.3 ± 2.9	2.1 ± 3.1	2.1 ± 3.1	2.1 ± 2.9	1.7 ± 2.8	<0.001
**Nasal obstruction** **(VAS)**	7.3 ± 2.5	3.9 ± 2.9	3.3 ± 2.6	2.4 ± 2.2	2.0 ± 2.1	2.2 ± 2.2	1.5 ± 1.6	1.8 ± 2.2	2.1 ± 1.7	1.7 ± 2.8	<0.001
**Rhinorrhoea** **(VAS)**	5.9 ± 2.6	3.7 ± 2.9	2.5 ± 2.3	1.8 ± 1.5	2.0 ± 2.4	1.8 ± 1.9	1.5 ± 1.7	1.9 ± 2.5	2.0 ± 1.8	1.8 ± 1.8	<0.001
**Sleep disorders** **(VAS)**	5.6 ± 2.6	2.7 ± 2.7	2.5 ± 2.4	2.2 ± 2.2	2.4 ± 2.3	2.0 ± 2.5	1.8 ± 2.3	1.8 ± 2.4	1.8 ± 2.2	1.7 ± 1.7	<0.001
**Craniofacial pain** **(VAS)**	4.9 ± 5.7	1.8 ± 2.6	2.0 ± 2.8	1.0 ± 1.9	1.2 ± 1.8	1.1 ± 1.7	0.9 ± 1.5	0.7 ± 1.9	1.2 ± 2	0.6 ± 1.5	<0.001
**Mean B-EOS** **(×10^9^/L)**	0.570 ± 0.323	-	0.803 ± 0.335	0.833 ± 0.504	0.840 ± 0.451	0.732 ± 0.494	0.581 ± 0.334	0.723 ± 0.629	0.704 ± 0.694	0.624 ± 0.576	NS
**Mean total IgE count** **(KU/L)**	167.4 ± 235.0	-	-	108.7 ± 186.3	39.8 ± 31.9	-	41.7 ± 88.4	35.3 ± 71.9	17.1 ± 13.5	17.7 ± 13.6	<0.05 **
**Lund–Mackay score** **(LMS)**	16.7 ± 5.1	-	-	-	7.4 ± 5.4	-	4.3 ± 5.1	-	8.2 ± 6.5	-	<0.001

SNOT-22: sinonasal outcome test-22; NPS: nasal polyp score; NCS: nasal congestion score; SSIT-16: Sniffin’s Sticks test; INCS: intranasal corticosteroids; VAS: visual analogue scale; B-EOS: absolute blood eosinophil count. VAS for sneezing, headache and post-nasal drip are not reported. Results of non-categorical variables are reported ad mean ± standard deviation. * *p* value comparing V0 to all the other time points (from V2 to V9); ** *p* value comparing V0 versus V4 to V7.

**Table 4 biomedicines-13-00501-t004:** Univariate correlations between continuous parameters at baseline.

		Total VAS	SNOT-22	NPS	B-EOS	Total IgE	LMS	Sniffin
**Total VAS**	R value *p* value	1.000						
**SNOT-22**	R value *p* value	0.526 0.004	1.000					
**NPS**	R value *p* value	−0.027 0.892	−0.287 0.124	1.000				
**B-EOS**	R value *p* value	−0.311 0.115	−0.043 0.825	−0.076 0.689	1.000			
**Total IgE**	R value *p* value	−0.222 0.286	−0.265 0.182	−0.259 0.192	0.375 0.059	1.000		
**LMS**	R value *p* value	0.009 0.963	−0.124 0.522	0.208 0.278	−0.039 0.840	0.275 0.174	1.000	
**SSIT-16**	R value *p* value	0.125 0.533	0.267 0.162	−0.350 0.058	−0.095 0.623	−0.100 0.626	−0.323 0.093	1.000

VAS: visual analogue scale; SNOT-22: sinonasal outcome test-22; NPS: nasal polyp score; B-EOS: absolute blood eosinophil count; LMS: Lund–Mackay score; SSIT-16: Sniffin’s Sticks test. Spearman’s or Pearson’s coefficients are reported as r and *p* values according to normality distribution of variables.

**Table 5 biomedicines-13-00501-t005:** Basal characteristics according to ACCESS score.

	ACCESS ≤ 10 (n = 14)	ACCESS > 10 (n = 11)	*p*
**Age**	61 ± 11	55 ± 10	0.217
**Sex (M)**	11 (78.6%)	9 (81.8)	0.762
**BMI**	25.1 ± 2.5	25.0 ± 2.9	0.805
**Atopy**	11 (28.6)	6 (54.5)	0.397
**NSAIDs intolerance**	4 (28.6%)	4 (36.4%)	0.986
**Presence of asthma**	11 (78.6%)	7 (63.6%)	0.706
**B-EOS (×10^9^/L)**	0.637 ± 0.043	0.511 ± 0.025	0.584
**Total IgE (KU/L) ***	126.9 ± 97.1	122.2 ± 110.9	0.733
**OCS needed in the last year**	12 (85.7%)	7 (63.6%)	0.417
**Prednisone in the last year (mg) ***	296.2 ± 467.8	283.4 ± 451.3	0.827
**OCS courses in the last year**	3.4 ± 3.2	1.4 ± 1.4	0.065
**Days of OCS in the last year ***	26.4 ± 23.8	70.7 ± 133.7	0.316
**n° of surgeries**	3.0 ± 1.9	2.1 ± 1.5	0.135
**Time between last surgery and biologics (months)**	58.8 ± 81.2	116.7 ± 79.3	0.026

Results of non-categorical variables are reported ad mean ± standard deviation. Categorical variables are reported as number (N) and percentage. B-EOS: absolute blood eosinophil count; NSAIDs: non-steroidal anti-inflammatory drugs; OCS: oral corticosteroids. Data available in * N = 13 and 10 respectively; Years and months are reported as mean and range.

**Table 6 biomedicines-13-00501-t006:** Mean paired delta value from V0 to V8 of total VAS, SNOT 22, NPS and Sniffin score, according to ACCESS score evaluation.

Delta	V0-V1	V0-V2	V0-V3	V0-V4	V0-V	V0-V6	V0-V7	V0-V8
Total VAS								
Access score ≤ 10	-	22.2 ± 16.0	26.7 ± 16.8	22.8 ± 16.7	-	30.1 ± 19.2	27.9 ± 21.2	19.6 ± 22.2
Access score > 10	-	20.0 ± 16.4	23.0 ± 12.8	23.6 ± 16.9	-	25.6 ± 16.3	23.4 ± 15.9	10.5 ± 23.3
*p* value	-	0.6208	0.9474	0.5992	-	0.7384	0.8688	0.7111
**SNOT-22**								
Access score ≤ 10	22.9 ± 21.3	24.3 ± 17.0	27.6 ± 20.3	29.5 ± 21.8	37.0 ± 19.8	32.3 ± 22.1	36.2 ± 27.8	22.8 ± 31.5
Access score > 10	22.3 ± 22.6	25.4 ± 22.2	34.4 ± 18.5	26.7 ± 19.3	24.7 ± 12.4	27.6 ± 17.3	32.2 ± 22.5	44.5 ± 2.1
*p* value	0.8693	0.9781	0.4268	0.7349	0.1132	0.8173	0.6790	0.8889
**NPS**								
Access score ≤ 10	1.4 ± 1.6	1.5 ± 1.6	2.3 ± 1.7	3.3 ± 1.9	3.8 ± 1.4	4.2 ± 1.5	3.7 ± 1.6	-
Access score > 10	1.6 ± 1.0	2.1 ± 1.6	2.4 ± 1.7	2.4 ± 1.6	3.2 ± 1.9	3.9 ± 2.0	4.6 ± 2.3	-
*p* value	0.3363	0.3000	1.0000	0.1983	0.4179	0.8931	0.3779	-
**SSIT-16**								
Access score ≤ 10	2.0 ± 3.2	2.8 ± 5.0	3.7 ± 4.5	3.2 ± 3.9	5.6 ± 5.0	5.5 ± 5.3	4.5 ± 6.0	5.3 ± 3.2
Access score > 10	3.2 ± 2.0	6.6 ± 3.7	4.4 ± 4.0	6.8 ± 4.3	6.8 ± 3.8	7.7 ± 4.7	6.9 ± 4.9	1.0 ± 2.8
*p* value	0.2198	**0.0658**	0.7238	**0.0736**	0.4665	0.3171	0.1974	0.1399

ACCESS score ≤ 10: n = 14; ACCESS score > 10: n = 11. Results of non-categorical variables are reported ad mean ± standard deviation. Statistics are not reported if data are missing or less than two paired data for groups. VAS: visual analogue scale; SNOT-22: sinonasal outcome test-22; NPS: nasal polyp score; SSIT-16: Sniffin’s Sticks test.

**Table 7 biomedicines-13-00501-t007:** Mean paired delta values from V0 to V8 of total VAS, SNOT 22, NPS and Sniffin score, according to number of previous surgeries (0 n = 4; 1 n = 8; 2 n = 7; ≥3 n = 13).

Paired Delta Value	V0–V1	V0–V2	V0–V3	V0–V4	V0–V5	V0–V6	V0–V7	V0–V8
Total VAS								
0	-	34.0 ± 10.0	28.0 ± 13.2	34 ± 7.8	-	36.7 ± 7.2	35.0 ± 9.5	-
1	-	10.9 ± 13.1	17.1 ± 14.3	12.0 ± 9.8	-	21.9 ± 13.4	24.3 ± 15.9	-
2	-	24.2 ± 14.0	29.0 ± 12.6	27.3 ± 14.2	-	32.2 ± 14.2	33.6 ± 12.4	-
≥3	-	26.9 ± 16.1	30.4 ± 15.4	25.2 ± 20.7	-	30.4 ± 25.0	21.5 ± 26.4	-
*p* value	-	0.0795	0.4907	0.0719		0.4315	0.6413	
**SNOT-22**								
0	32.0 ± 18.6	41.7 ± 20.6	37.0 ± 20.2	38.5 ± 15.3	38.0 ± 19.1	36 ± 18.6	41.3 ± 21.9	35.5 ± 27.6
1	12.4 ± 12.3	15.7 ± 9.2	19.6 ± 15.2	13.7 ± 14.1	24.0 ± 13.7	18.8 ± 16.6	27.5 ± 22.8	19.0 ± 38.2
2	15.5 ± 18.6	24.0 ± 16.8	32.0 ± 14.9	31.6 ± 15.6	37.1 ± 14.7	37.2 ± 14.0	41.5 ± 14.3	42.0 ± 21.5
≥3	31.6 ± 23.6	26.9 ± 21.1	32.3 ± 21.5	34.6 ± 22.6	31.8 ± 21.1	35.8 ± 21.4	34.4 ± 32.0	21.0 ± 29.1
*p* value	0.1018	0.2410	0.5697	0.1183	0.4847	0.3117	0.7872	0.4790
**NPS**								
0	1.3 ± 1.3	2.3 ± 1.2	2.8 ± 2.2	3.8 ± 2.2	4.3 ± 2.1	5.3 ± 1.3	4.7 ± 1.5	4.5 ± 0.7
1	1.3 ± 0.7	1.1 ± 0.7	2.0 ± 1.6	2.3 ± 1.5	3.1 ± 2.0	3.8 ± 1.8	3.5 ± 1.6	4.0 ± 2.8
2	1.7 ± 1.2	2.8 ± 1.6	3.2 ± 1.7	3.3 ± 1.9	3.8 ± 1.6	4.3 ± 2.0	4.3 ± 2.1	5.3 ± 1.2
≥3	1.3 ± 1.2	1.4 ± 1.4	2.0 ± 1.4	3.1 ± 1.9	3.7 ± 1.5	4.0 ± 1.4	4.4 ± 2.0	3.4 ± 2.3
*p* value	0.8898	0.1425	0.4542	0.6069	0.8213	0.4476	0.7291	0.4351
SSIT-16								
0	6.5 ± 7.0	8.7 ± 7.6	3.8 ± 13.0	8.5 ± 6.5	9.5 ± 6.5	8.5 ± 6.0	9.0 ± 7.0	8.0 ± 11.3
1	3.0 ± 1.3	5.3 ± 5.7	6.6 ± 4.2	5.9 ± 5.3	6.6 ± 4.9	8.1 ± 5.0	6.3 ± 6.4	1.5 ± 3.5
2	1.8 ± 2.9	6.7 ± 4.2	3.0 ± 3.3	4.3 ± 5.1	7.1 ± 4.7	8.0 ± 6.3	5.9 ± 8.5	8.7 ± 10.9
≥3	2.6 ± 3.4	2.3 ± 4.0	3.3 ± 4.3	4.3 ± 3.5	4.7 ± 3.9	3.8 ± 3.2	3.1 ± 4.2	4.5 ± 6.0
*p* value	0.7375	0.1050	0.3951	0.5835	0.4149	0.2165	0.4066	0.5531

Number of previous surgeries 0: n = 4; 1: n = 8; 2: n = 7; ≥3: n = 13. Statistics are not reported if data are missing or less than two paired data for groups. VAS: visual analogue scale; SNOT-22: sinonasal outcome test-22; NPS: nasal polyp score; SSIT-16: Sniffin’s Sticks test.

## Data Availability

The original contributions presented in this study are included in the article. Further inquiries can be directed to the corresponding author.

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
