# Peer review of "Dupilumab in the Treatment of Severe Uncontrolled Chronic Rhinosinusitis with Nasal Polyps (CRSwNP) and Comorbid Asthma—A Multidisciplinary Monocentric Real-Life Study"

_biomedicines, 2025, doi:10.3390/biomedicines13020501_

Round 1
Reviewer 1 Report
Comments and Suggestions for Authors
Suggestions:
In methods section, exclusion criteria are clear but the exclusion criteria should be further clarified. Data has been treated with proper statistical methods and the results are presented nicely. Discussions section is adequate but can be improved both in Language and also in Discussion as I suggested below.
For instance:
Some sentences are a bit vague: "We suppose in the future patients will start biologics sooner, without needing multiple re-interventions and there will be an increase in the number of naive patients". I suggest author can clarify this.
Lines 406-409: Authors state that "In our experience, no statistical correlation was found be- 406 tween ACCESS score and clinical response, contrary to a study by Gian Marco Pace et al 407 [26] where ACCESS and response to therapy were correlated in patients treated for at least 408 12 months.". Please provide a possible explanation for this discrepancy, and also briefly suggest a solution to find out how this can be solved in future studies.
Author Response
Reviewer 1
MINOR ISSUES
Comment 1: In methods section, exclusion criteria are clear but the exclusion criteria should be further clarified
Response1 : We thank the Reviewer for the comment. We added exclusion criteria in the text, Section Materials and Methods, subsection Patients and study design, page3, line 94-96.
Comment 2: . Data has been treated with proper statistical methods and the results are presented nicely. Discussions section is adequate but can be improved both in Language and also in Discussion as I suggested below. For instance: Some sentences are a bit vague: "We suppose in the future patients will start biologics sooner, without needing multiple re-interventions and there will be an increase in the number of naive patients". I suggest author can clarify this.
Response 2: We thank the Reviewer for the comment. Were reformulated the sentence as follow: ”We suppose in the future patients will start biologics before undergoing multiple uncompleted surgeries, due to the improvement of surgical expertise as well as to the acquisition of new knowledge about the effectiveness of biologics. (page 17, line 437-439)
Comment 3: Lines 406-409: Authors state that "In our experience, no statistical correlation was found between ACCESS score and clinical response, contrary to a study by Gian Marco Pace et al [26] where ACCESS and response to therapy were correlated in patients treated for at least 12 months.". Please provide a possible explanation for this discrepancy, and also briefly suggest a solution to find out how this can be solved in future studies.
Response 3 : We thank the Reviewer for the comment. We reformulated the sentence in a more comprehensive form (page 17, line 441-449). We also added a reference (new REF 32) concerning the suggested explanation. In addition, we modified in materials and methods section the ACCESS score cut-off definition (page 4, line 131).

Reviewer 2 Report
Comments and Suggestions for Authors
In this manuscript, the author explored the impact of Dupilumab on clinical outcomes and quality of life in patients affected by both asthma and CRSwNP. This study lasts over 30 months and data from the study is crucial which points out the optimal strategy of using dupilumab in the treatment on certain types of CRSwNP. Have a few comments/questions/suggestions:
Line 72-73 please provide the detailed introduction of SNOT22, VAS, and NCS. Not clear how the score is evaluated, how severe is the symptom based on the score.
Line 121-122. Please describe how B-EOS was measured, and total serum ige level, by elisa?
Table 1,2 3,5, and 6. Are the data in the tables mean ± SD or SEM?
Table 4. why B-eos and total ige are related to the other scoring system in a reverse way? Would expect inflammation biomarkers such as eosinophil and ige are positive related with the severity of the symptom?
Figure 1,2,4. Please provide the info of what stats analysis were ran for each figure in the legend.
Figure 2. add the units in the Y axis.
Figure 2b. v6 not statistically significant?
Figure 2c. v8, v9 not statistically significant?
Figure 3B, Panel C and D. Either define ‘ss’ and ‘sm’ or remove them from the figure. Please adjust the size of the endoscopic image when comparing the efficacy of the treatment. At least in the same panel, like in figure 3b, panel d, unify the size of images.
Author Response
Reviewer 2
MINOR ISSUES
Comment 1: Line 72-73 please provide the detailed introduction of SNOT22, VAS, and NCS. Not clear how the score is evaluated, how severe is the symptom based on the score
Response 1: We thank the Reviewer for the comment. We added in the Materials and Methods section the range of evaluation scale for total (range 0-80) and single domain (range 0-10) VAS score, SNOT-22 (range 0–110) and NCS (range 0-3) and a more detailed explanation about their evaluation. (page 3, line 117-124). We added a reference concerning this topic. (new REF 13)
Comment 2: Line 121-122. Please describe how B-EOS was measured, and total serum ige level, by elisa?
Response 2: We thank the Reviewer for the comment. B-EOS were counted by automated analysis and expressed in cells per microliter (cells/mcl). Unit of measure has been reported in all tables and text for clarification. Total IgE levels were measured by enzyme-linked immunosorbent assay (Elisa) and expressed in KU/L. clarification has been added in the text (page 4, line 134)
Comment 3: Table 1,2 3,5, and 6. Are the data in the tables mean ± SD or SEM?
Response 2: We thank the Reviewer for the comment. As reported in each table caption, results of non categorical variables are reported ad mean ± standard deviation. Specific caption has been added in table 6.
Comment 4: Table 4. why B-eos and total ige are related to the other scoring system in a reverse way? Would expect inflammation biomarkers such as eosinophil and ige are positive related with the severity of the symptom?
Response 4 : We thank the Reviewer for the comment. This is a very interesting observation. In table 4 we did not report any significant correlation between B-EOS and total IgE compared to symptoms and endoscopic scores. Although some studies reported an association between symptoms and increased B-EOS, the major role for this biomarker is in distinguishing CRSwNP phenotypes as well as predicting polyp recurrence. This point has been addressed in the discussion and a new reference have been added (new REF 21), page 16, line 380-384. Indeed the search for local biomarkers that can better characterize the spectrum of severity of CRSwNP compared to B-EOS is still ongoing, therefore not surprising the negative correlation (r) , although not significant, among B-EOS and symptoms ( new REF 22) page 16, line 384-387. Total IgE are generally considered to have low sensitivity in predicting type-2 inflammation, as demonstrated in asthma (new REF 23), and their role in CRSwNP is more related to the interaction with specific triggers (Aspirin, molds, staphylococcus endotoxins) that trigger the type-2 pathway. This point has been added in the discussion, page 17, line 391-94 (new REF 24).
Comment 5: Figure 1,2,4. Please provide the info of what stats analysis were ran for each figure in the legend.
Response 5 : We thank the Reviewer for the comment. Stats analysis in figure legends have been added
Comment 6: Figure 2. add the units in the Y axis. Response 6: Units have been added in the legend.
Comment 7: Figure 2b. v6 not statistically significant? . Response 7: V6 compared to V0 is not statistically significant (P= 0.8308)
Comment 8 Figure 2c. v8, v9 not statistically significant? Response8: V8 compared to V0 is statistically significant (p=0.0127), accordingly we changed the result both in table 3 an Figure 2C . V8 compared to V0 is statistically significant (p= n 0,058)
Comment 9 : Figure 3B, Panel C and D. Either define ‘ss’ and ‘sm’ or remove them from the figure. Response 9: “ss”sphenoid sinus; “sm” maxillary sinus. Definition have been added in Figure3, legend.
Comment 10 Please adjust the size of the endoscopic image when comparing the efficacy of the treatment. At least in the same panel, like in figure 3b, panel d, unify the size of images.
Response 10: Figure3b. Endoscopic images of pane A,C,D,E have been revised. (Left and Right). Resizing was not possible as these are camera original Photos. However, though the comparison of left and right images, over time treatment effect is observed.
